# Breastfeeding self-efficacy status and associated factors among postpartum mothers at Hadiya Zone public hospitals, Southern Ethiopia

**Lemlem Nigussiee[1]\*, Tigist Demeke[2], Vinod Bagilkar[3], Yemisrach Firetsidk[1], Mulatu Abageda** [1,4]**, Tefera Belachew[5]**

**1** Department of midwifery, College of Medicine and Health Sciences, Wachemo University, Hosanna, Ethiopia, **2** Jimma University, Institute of Health, School of Nursing, Jimma, Ethiopia, **3** Department of Pediatric Nursing, Crescent college of nursing Belagavi, Karnataka, India, **4** Department of Population and Family Health, Jimma University, Institute of Health, Jimma, Ethiopia, **5** Department of nutrition and dietetics, Jimma University, Institute of Health, Jimma, Ethiopia

\* lemlemnegussie10@gmail.com

## Abstract

### Background

Breastfeeding is a crucial health-promoting behavior that has several positive effects on the health of both mothers and newborns. Breastfeeding self-efficacy (BFSE) status is an important factor that is positively related to breastfeeding, as it influences a woman's decision to breastfeed or not. Therefore, the objective of this study is to assess the BFSE status and its associated factors among postpartum mothers at public hospitals in the Hadiya Zone.

### Method

A cross-sectional study design was conducted from August 1–30, 2022, at public hospitals in the Hadiya Zone, southern Ethiopia. Data were collected from 416 participants using a pretested, structured questionnaire. Study participants were selected using a systematic random sampling technique from four public hospitals. The data were entered into Epi Data version 3.1 and analyzed using the SPSS version 24. Bivariable analysis was conducted initially, and variables with p values ≤0.25 were entered into multivariable logistic regression. Statistical significance was declared at a p value ≤0.05.

### Results

The results of this study showed that 48% had a high level of BFSE. On multivariable logistic regression analysis, the variables age ≥35 years old [AOR = 3.596; 95% CI (1.564, 8.26)], age 20 to 34 [AOR = 2.352; 95% CI (1.2275, 4.506)], spontaneous vaginal delivery [AOR = 2.755; 95% CI (1.636, 5.566)], breastfeeding experience [AOR =1.845; 95% CI (1.028, 3.3)], more than secondary educational status [AOR = 6.856; 95% CI

**Data availability statement:** All relevant data is within the manuscript and its Supporting information files.

**Funding:** This study received a student research fund from Jimma University. The funder has not played any role in the study design, collection of data, or development of this manuscript.

**Competing interests:** The authors have declared that no competing interests exist.

**Abbreviations:** ANC, Antenatal care; AOR, Adjusted Odds Ratio; BF, Breastfeeding; BFSE, Breastfeeding self-efficacy; BFSE-SF, Breastfeeding self-efficacy short form; C/S, Cesarean section; NGOs, Nongovernmental organizations; OR, Odds Ratio; SVD, Spontaneous vaginal delivery; UNICEF, United Nations International Children's Emergency Fund; WHO, World Health Organization

(2.670–17.603)], and intended pregnancy [AOR = 4.156; 95% CI (2.239, 7.714)] were significantly associated with high breastfeeding self-efficacy status.

## Conclusion

The study findings suggest that mothers attending postpartum care at Hadiya Zone Hospital have a low status of breastfeeding self-efficacy. Therefore, efforts should be made to enhance mothers' breastfeeding self-efficacy status before they are discharged from the hospital.

## Background

Breastfeeding self-efficacy (BFSE) refers to a mother's perception or confidence in her ability to breastfeed her baby, rather than her actual practice [1]. In the early postpartum period, a mother's BFSE status can be used to identify mothers who are more likely to be successful in breastfeeding (BF) and those who need further support to ensure BF. Therefore, BF initiation, length, and exclusivity rates among mothers in the postpartum period can be predicted by maternal BFSE status [2].

Furthermore, an increase in BF self-efficacy from medium to high reduces the likelihood of BF discontinuation by 80%, while a shift from low to medium reduces it by 48% [3–5]. This highlights the significance of having a high level of self-efficacy for successful BF, as well as the potential for failure associated with low self-efficacy. Consequently, mothers with higher BFSE are more likely to initiate BF and adhere to it, whereas those with lower self-efficacy may struggle to even start BF [6,7].

In a broader context, universal BF could prevent 20,000 maternal deaths from breast cancer and 823,000 deaths of under five children worldwide each year if it were practiced universally [8]. However, based on data from health surveillance in 130 countries, the lack of BF resulted in an additional 166 million cases of diarrheal illnesses and 9 million cases of pneumonia in children under the age of two. Among these cases, 84% occurred in low- and middle-income countries, with two-thirds of them concentrated in South Asia and Sub-Saharan Africa. Apart from the physiological consequences, the absence of BF carries economic implications. For instance, the annual costs of unnecessary treatments for childhood pneumonia and diarrhea incurred by healthcare systems in North America, Southeast Asia, and Sub-Saharan Africa due to the lack of BF are estimated at $1.83 million, $133.8 million, and $152.4 million, respectively [8,9].

To address this burdensome issue and increase BF rates, international organizations like the United Nations International Children's Emergency Fund (UNICEF) and the World Health Organization (WHO) have implemented various initiatives and programs aimed at enhancing BF practices through healthcare providers and institutions [10]. Incentives have also been introduced to promote BF, such as paid BF breaks, workplace policies supporting breastfeeding, and the "World Breastfeeding Week" campaign [11,12] Despite these efforts, the desired level of BF has not yet been achieved [13].

Thus far, prior to engaging in a particular behavior, there should be a high level of self-efficacy for that behavior. If self-efficacy is lacking, individuals tend to behave ineffectively, despite knowing what to do, as stated by Albert Bandura [14]. Specifically, mothers with a high status of breastfeeding self-efficacy (BFSE) can continue BF even in unfavorable situations and are self-motivated. Conversely, this is not the case for those with low-status BFSE. This illustrates how BFSE status can influence both the initiation and continuation of

breastfeeding. Therefore, BFSE is a crucial variable and a determining factor for the initiation, success, and maintenance of breastfeeding simultaneously [1,15].

Furthermore, in order to achieve the sustainable development goal by 2030, the importance of BF is clearly evident and aligns with targets 2.1, 3.1, and 4.4. However, according to trends among 113 countries from 2000–2019, there has been an increase in the global rate of exclusive BF, but it is still insufficient to reach the goal solely by promoting exclusive BF [16]. Similarly, based on the Ethiopian demographic health survey, exclusive BF increased from 49 to 59 percent among children under the age of six months. However, BF among 12- to 18-month-old children decreased from 85 to 76%. This decline in BF may be attributed to the low status of BFSE among mothers [17,18].

Finally, attention to modifiable variables that can assess the mother herself on BF, such as maternal BFSE, is imperative. However, a systematic review on BF and related factors in Ethiopia discovered that variables frequently studied included antenatal care, postpartum care, institutional delivery, maternal occupation, husband support, family income, age, health service-related factors, marital status, and parity [17]. To the best of our knowledge, there has been no research conducted on BFSE and associated factors among postpartum mothers in Ethiopia, as well as in the study area. Therefore, the aim of this study is to assess the status of BFSE and its associated factors among postpartum mothers in Hadiya zone public hospitals, southern Ethiopia.

## Method

### Study setting

The research was carried out in all of the public hospitals in the Hadiya Zone. The Hadiya Zone is one of the zones in the southern part of Ethiopia. Hadiya Zone has 13 districts and four town administrations. According to the annual report of the zone's health department, the total population is 1,688,820, of whom 50.44% are women. The total reproductive age groups were 39,493 (23.3%), and the estimated pregnancy was 53,367. The Zone has four hospitals (3 general and 1 specialized comprehensive hospital), 59 health centers, and 311 health posts, all of which are government-run [19]. The study was conducted from August 1–30, 2022.

### Study design

Institution based cross-sectional study design was conducted.

### Inclusion and exclusion criteria

**Inclusion.** Postpartum mothers who gave birth in all of the hospitals and who gave birth either by SVD or C/S during the data collection period and willing to participate in the study were included.

**Exclusion.** Critically ill mothers were excluded.

**Sampling method.** The sample size of the study was determined by the single population proportion formula. Considering p = 50%. The following parameter was used to calculate the sample size.

$$n = \frac{(z\alpha/2)^2 \, p(1-P)}{d^2}$$

where n is the maximum possible sample size, Z α/2 is the 95% confidence interval = 1.96, P is the population proportion = 0.5%, and d is the margin of error with a 95% confidence interval = 0.05.

$$n = \frac{(1.96)^2 * 0.5(1-0.5)}{0.05^2} = 384$$

Considering the non-response rate 10% which was (~ 38), was added this gives the final sample size 422. All four (100%) of the governmental Hadiya zone hospitals were included in this study. The average number of monthly postpartum clients from Wachemo University Neigist Elleni Mohamed Memorial Comprehensive Specialized Hospital (WUNEMMCSH), Shone General Hospital, Homacho General Hospital, and Gimbichu General Hospital is 764, 234, 193, and 132, respectively. The sample was taken by proportional allocation from each hospital. A systematic random sampling technique was used to select study participants. Three was the Kth value used to select participants, determined by dividing the total number of postpartum women by the sample size (1323/422 = 3). All postpartum mothers who were in the postpartum and maternity wards were selected, while mothers who gave birth by cesarean section were interviewed after 8 hours postpartum. The lottery method was used to select the first participant from the first three clients. Finally, the other participants were selected at every 3rd interval until the total sample size was reached.

## Data collection methods and outcome measurements

The data were collected using a pretested, structured, interviewer-administered questionnaire. The questionnaire was adapted from previous relevant literature [18,20–22] and the questions were closed-ended. The data was collected by four midwives with BSc degree under the supervision of one midwife with MSc. The questionnaire was designed to obtain information on the BFSE status of postpartum mothers. On the other hand, sociodemographic, obstetric, and health service-related questions are utilized by the questionnaire. To measure maternal BFSE status at postpartum time, Breastfeeding Self-efficacy scale – short-form (BSES-SF) was used which was developed by Cindy-Lee Dennis [23]. It has a Crombach alpha value (α = 0.97) on the original scale. The breastfeeding self-efficacy short form (BFSE-SF), a set of 14 Likert scale questions, was used to evaluate the BFSE status of postpartum mothers. The response was considered as 1 = not at all confident to 5 = very confident. The total score was in the range of 14 to 70 [23]. After computing the variables, it was dichotomized by considering the median value of the responses, which was stated as having a high status of BFSE for the score ≥ median value, which was 57, and below the median, which was considered to have a low status of BFSE [18]. In this study the crombach alpha of BFSE-S is 0.96.

## Data quality control

The original Breastfeeding Self-efficacy scale – short-form (BSES-SF was translated into Amharic (local language) and then back to English by experts of the language to keep its consistency. Training on data collection tools was given to data collectors and supervisors. The data was collected by four midwives with BSc degree and previous data collection experience under the supervision of one midwife with MSc. Pre-testing of the questioner was done on 5% of the questioner outside the research zone (Halaba General Hospital) before the study was conducted, and the appropriate corrections were made.

## Data analysis

The data were edited, coded, cleaned, and entered into Epi Data version 3.1 before being exported to the statistical software SPSS version 24.0 for analysis. The study variables were described using various frequency tables, graphs, and descriptive summaries. There is no multicollinearity among the independent variables. Bivariable logistic regression analysis was used

to see a significant association between the outcome and independent variables. All explanatory variable which have association in bivariate analysis with p value less than or equal 0.25 were entered in to multiple logistic regression model in order to assess the independent predictors of postpartum BFSE status. A multivariable logistic regression was performed to identify the independent predictor variables of postpartum BFSE status. $P \leq .05$ was considered statistically significant in the study. The strength of the association between the outcome and the explanatory variables was measured by using odds ratios with a 95% confidence interval.

### Ethical consideration

Ethical clearance was obtained from Jimma University's institutional review board (IRB) of the Institute of Health by reference number JUIRB/P4/22. The letter was sent to the Hadiya zone hospitals, and they permitted it. All participants were informed about the study's objectives, and written informed consent was obtained. All data were examined by the lead investigator, and confidentiality and privacy were maintained throughout the study procedure.

## Result

### Socio-demographic factors

The planned sample size was 422, and the total number of study subjects involved was 416, with a response rate of 98.58%. The mean age of the mothers was 28.82 years. More than half (61.5%) of the mothers were aged 25–34 years. Regarding the mother's educational status, 109 (26.5%) and 157 (37.7%) of the mothers attended primary (1–8) and secondary (9–12) school, respectively. The educational status of the husbands was 293 (70.4%) of them were secondary or above (Table 1).

### Obstetric and health service-related factors

Of the total study participants, more than half (75%) were multiparous and 317 (76.2%) respondents visited health institutions during pregnancy. Four hundred sixteen (100%) of the study participants planned to breastfeed their newborns. Regarding the desire for pregnancy status, 296 (71.2%) was intended. Concerning the mode of delivery, 347 (84.3%) were spontaneous vaginal deliveries (Table 2).

### Breastfeeding self-efficacy status of the postpartum mothers

From the total questions, item number 6 on the scale, which is called I can manage to breastfeed even if my baby is crying, was the leading item, which was 35.8%. Item number 14, called I can always tell when my baby is finished BF, had the lowest response among respondents, which was 26.4% in the BFSE-SF (Table 3).

### Bivariate and multivariable logistic regression analysis of factors predicting breastfeeding self-efficacy status among postpartum mothers

Bivariable analysis in the binary logistic regression model shows that age of the mother, educational status, marital status, wealth index, pregnancy status, mode of delivery, antenatal care visit, frequency of ANC visit, parity, and breastfeeding experience were factors affecting postpartum mothers' BFSE at $P < 0.25$. In the multivariable analysis, adjusting for possible confounding variables, maternal age, educational status, mode of delivery, BF experience, and pregnancy status were statistically associated with BFSE at $P < 0.05$.

In multivariable analysis, mothers in the age group 25–34 were two times [AOR = 2.352, 95% CI (1.227, 4.506)] and those in the age group of 35 years and older were three times [AOR = 3.596,

**Table 1. Sociodemographic characteristics of the respondents in Hadiya Zone public hospitals, Southern Ethiopia 2022.**

| Characteristics | Category | Frequency (N) | Percent (%) |
|---|---|---|---|
| Age of mother | < 20 | 79 | 19 |
| | 20–34 | 256 | 61.5 |
| | ≥ 35 | 81 | 19.5 |
| Educational status of the mother | No education | 89 | 21.4 |
| | Primary | 109 | 26.2 |
| | Secondary | 157 | 37.7 |
| | More than secondary | 61 | 14.7 |
| Marital status | Never married | 17 | 4.1 |
| | Married | 375 | 90.1 |
| | Divorced | 24 | 5.8 |
| Mother's Occupation | Government employ | 117 | 28.1 |
| | Self- employ | 163 | 39.2 |
| | Housewives | 136 | 32.7 |
| Husband education | No education | 31 | 7.4 |
| | Primary | 92 | 22.1 |
| | Secondary | 197 | 47.5 |
| | More than secondary | 96 | 23 |
| Residence | Urban | 195 | 46.9 |
| | Rural | 221 | 53.1 |
| Religion | Orthodox | 109 | 26.2 |
| | Protestant | 219 | 52.6 |
| | Muslim | 60 | 14.4 |
| | Other | 28 | 6.8 |
| Wealth status | Poorest | 83 | 20 |
| | Poorer | 85 | 20.4 |
| | Middle | 82 | 19.7 |
| | Richer | 83 | 20 |
| | Richest | 83 | 20 |

95% CI (1.564, 8.268)] more likely to have high BFSE than those who were less than 25 years old. Mothers who gave birth by spontaneous vaginal delivery were two times [AOR = 2.755, 95% CI (1.636, 5.566)] more likely to have a high BFSE than those who gave birth by cesarean section. Mothers with more than a secondary education were found to be six times [AOR = 6.86, 95% CI (2.670, 17.603)] more likely to have high BFSE scores compared to mothers without any formal education. Participants who had previous BF experience were 1.8 times [AOR =1.85 (1.028, 3.313)] more likely to have a high BFSE status compared to those without experience (Table 4).

## Discussion

The breastfeeding practices of mothers are influenced by a range of factors, including BFSE status, socio-demographic variables, and past experiences. Despite the widespread acceptance and celebration of BF in Ethiopian culture, mothers do not consistently adhere to optimal BF. As far as we are aware, no previous studies have employed validated questionnaires to assess BFSE in Ethiopia, particularly in our study area. The findings of the current study can assist health professionals and decision-makers in designing and implementing culturally tailored interventions to enhance maternal breastfeeding self-efficacy.

**Table 2. Breast feeding self-efficacy status of postpartum mothers scored by item in Hadiya Zone public hospitals in southern Ethiopia 2022 (n = 416).**

| Item | Category (%) | | | | |
|---|---|---|---|---|---|
| | 1 | 2 | 3 | 4 | 5 |
| I can successfully cope with breastfeeding like I have with other challenging tasks. | 66 (15.9) | 62 (14.9) | 5 (1.2) | 143 (34.4) | 140 (33.7) |
| I can Breastfeed my baby without using additional thing before six months | 79 (19.0) | 69 (16.6) | 7 (1.7) | 125 (30.0) | 136 (32.7) |
| I can ensure that my baby is properly latched on for the whole feeding. | 79 (19) | 61 (14.7) | 4 (1) | 144 (34.6) | 128 (30.8) |
| I can always manage the breastfeeding situation to my satisfaction | 75 (18.0) | 52 (12.5) | 9 (2.2) | 148 (35.6) | 132 (31.7) |
| I can always determine that my baby is getting enough milk | 84 (20.2) | 66 (15.9) | 8 (1.9) | 131 (31.5) | 127 (30.5) |
| I can always manage to breastfeed even if my baby is crying | 73 (17.5) | 76 (18.3) | 8 (1.9) | 110 (26.4) | 149 (35.8) |
| I can always Keep wanting to breastfeed | 65 (15.6) | 66 (15.9) | 11 (2.6) | 127 (30.5) | 147 (35.3) |
| I can always comfortably breastfeed with my family members present | 62 (14.9) | 86 (20.7) | 9 (2.2) | 117 (28.1) | 142 (34.1) |
| I can always be satisfied with my breastfeeding. | 75 (18) | 70 (16.8) | 9 (2.2) | 134 (32.2) | 128 (30.8) |
| I can always deal with the fact that breastfeeding can be time-consuming | 65 (15.6) | 69 (16.6) | 11 (2.6) | 132 (31.7) | 139 (33.4) |
| I can Finish feeding my baby on one breast before switching to the other breastfeeding | 81 (19.5) | 71 (17.1) | 7 (1.7) | 133 (32.0) | 124 (29.8) |
| I can continue to breastfeed my baby for every feeding | 72 (17.3) | 69 (16.6) | 9 (2.2) | 134 (32.2) | 132 (31.7) |
| I can always manage to keep up with my baby's breastfeeding demands | 86 (20.7) | 63 (15.1) | 7 (1.7) | 131 (31.5) | 129 (31.0) |
| I can tell when my baby is finished breastfeeding | 112 (26.9) | 67 (16.1) | 6 (1.4) | 121 (29.1) | 110 (26.4) |

1(Not at all confident) 2(Sometimes confident) 3(Neutral) 4(Confident) 5(Very confident)

**Table 3. Distribution of the study participants by obstetric characteristics and health service of postpartum mothers at Hadiya Zone public hospitals, southern Ethiopia 2022 (n = 416).**

| Variable | Category | frequency | Percent (%) |
|---|---|---|---|
| Parity | Prim Para | 104 | 25 |
| | Multi Para | 312 | 75 |
| ANC visit | Yes | 317 | 76.2 |
| | No | 99 | 23.8 |
| ANC frequency(N = 317) | 1–3 | 144 | 45.5 |
| | ≥ 4 | 173 | 54.5 |
| Learn on breast feeding at ANC visit(N = 317) | Yes | 223 | 68.5 |
| | No | 94 | 31.4 |
| Current pregnancy status | Intended | 296 | 71.2 |
| | Un intended | 120 | 28.8 |
| Gestational age at birth | Preterm | 15 | 3.6 |
| | Post term | 12 | 2.9 |
| | Term | 389 | 93.5 |
| Breastfeeding experience | Yes | 310 | 74.5 |
| | No | 106 | 25.5 |
| Do you have a plan to breastfeeding | yes | 416 | 100 |
| Mode of de Livery | SVD | 347 | 84.3 |
| | CS | 69 | 16.6 |
| Sex of current baby | Male | 218 | 52.4 |
| | Female | 198 | 47.6 |

The present study revealed a notable finding in relation to BFSE. The overall prevalence of high BFSE was found to be 48%, with a 95% CI (43.7–47.5). The findings show that breastfeeding mothers in the study area do not have greater confidence in breastfeeding

**Table 4. Bivariate and multivariable logistic regression analysis of predictive variables of breastfeeding self-efficacy status among postpartum mothers in Hadiya Zone public hospitals in southern Ethiopia 2022.**

| Variables | Category | BFSE status | | COR 95%CI | AOR 95%CI | P value |
|---|---|---|---|---|---|---|
| | | High | low | | | |
| Mother age | < 20 | 23 | 56 | 1 | 1 | |
| | 20–34 | 130 | 126 | 2.512 (1.458, 4.327) | 2.352 (1.227, 4.506) | 0.010 |
| | ≥ 35 | 49 | 32 | 3.728 (1.929, 7.205) | 3.596 (1.564, 8.268) | 0.003 |
| Mothers educational status | No formal education | 33 | 56 | 1 | 1 | |
| | Primary | 55 | 54 | 1.728 (0.977, 3.059) | 1.365 (0.711, 2.621) | 0.350 |
| | Secondary | 67 | 90 | 1.263 (0.741, 2.155) | 1.286 (0.681, 2.428) | 0.438 |
| | More than secondary | 47 | 14 | 5.697 (2.730, 11.888) | 6.856 (2.670, 17.603) | 0.001 |
| Mode of delivery | Spontaneous vaginal delivery | 181 | 166 | 2.492 (1.432, 4.339) | 2.755 (1.363, 5.566) | 0. 005 |
| | Caesarian section | 21 | 48 | 1 | 1 | |
| Breastfeeding experience | Yes | 167 | 143 | 2.369 (1.492, 3.761) | 1.845 (1.028, 3.313) | 0.040 |
| | No | 35 | 71 | 1 | 1 | |
| Pregnancy status | Intended | 167 | 129 | 3.144 (1.993, 4.959) | 4.156 (2.239, 7.714) | 0.001 |
| | unintended | 35 | 85 | 1 | 1 | |

their infants. This percentage surpasses previous research conducted in Brazil, where only 35% of mothers exhibited high BFSE [24], An explanation for this discrepancy may be the fact that out of all mothers who participated in our study, 68.5% adhered to advice from medical professionals on BF. Additionally, a study from China reported that 43.9% of postpartum mothers had a high BFSE status. This variability can likely be attributed to the fact that 74.5% of the present study participants had previous experience with breastfeeding. It is plausible that the absence of BF experience and inadequate education during antenatal care visits contribute to lower levels of BFSE among mothers [25]. Furthermore, differences in economic and cultural factors between countries may also contribute to variations in BFSE status.

Conversely, this study found that mothers' BFSE status was significantly lower compared to a study conducted in Nepal, where it was 94% [26]. This variation may be attributable to the fact that Nepalis sample consisted of mothers who breastfed continuously for at least two weeks. It is known that mothers in the late postpartum period, which is more than 24 hours after delivery, exhibit greater stability in their physical, physiological, and psychological domains compared to the participants in present study who were in immediate postpartum period. Consequently, this indicates that mothers in the late postpartum period may have a higher BFSE status. Additionally, in Nepal, 95% of the study participants reported receiving support from their husbands, which also increases the probability of having a higher BFSE status [26,27].

Furthermore, in Brazil, 95.3% of postpartum mothers have a high BFSE status [28]. One potential explanation for this inconsistency could be that preterm mothers were excluded from the research, providing an opportunity for a higher BFSE status. This is supported by evidence that mothers of preterm babies face greater challenges due to prematurity-related issues, which can lead to physiological responses such as fatigue, stress, and anxiety, all of which can impact BFSE [29,30]. Additionally, more than half (57%) of the participants in Vientiane, Laos, Asia, had a high BFSE status. This could be attributed to the fact that 95% of the participants attended antenatal care sessions more than four times. This implies that having ANC follow-up more than four times increases the chances of mothers possessing a high BFSE status [21].

In our findings, age was significantly associated with maternal BFSE status. Mothers in the age range of 20 to 24 were twice as likely to have high BFSE compared to mothers under 20 years old, while mothers aged 35 and above were three times more likely to have high BFSE compared to the same reference groups. This indicates that older mothers are more likely to have high BFSE. This result aligns with other research conducted in Turkey and Vietnam [29,31]. One possible explanation for this association could be that older women are more likely to possess a high BFSE status [29].

A significant association was discovered between educational status and breastfeeding BFSE. Mothers with more than a secondary education were found to be six times more likely to have high BFSE scores compared to mothers without any formal education. This finding is consistent with research conducted in western Indonesia [18]. It is likely that individuals with higher education have better access to health services and information about BF from various relevant sources, which enables them to have a greater understanding of the benefits of BF and better prepare themselves for BF from all perspectives, resulting in a higher BFSE status.

In terms of the mode of delivery in this study, the findings indicate a significant association with postpartum mothers' BFSE. Mothers who gave birth through spontaneous vaginal delivery (SVD) were twice as likely to have a high BFSE status compared to those who had a cesarean section (CS). This could be because mothers who give birth through SVD require less intensive care and experience less physical discomfort, which may contribute to a higher BFSE status compared to those who had a CS birth [18,32]. This result aligns with a previous investigation conducted in Uganda [20]. However, in a study conducted in Tehran, Iran, having a CS birth was not found to be linked with BFSE. This discrepancy could be to the fact that out of all the study participants, 72% had CS births, and among those, 65.3% were planned [33]. The mother's planned mode of delivery might have influenced her preparation for the procedure and her psychological outlook, which in turn could have an impact on BFSE.

Regarding the experience of BF, this study found that participants who had previous experience with BF were 1.8times more likely to have a high BFSE status compared to those without experience. This finding is consistent with studies conducted in virous countries, including the United Kingdom, Goiania/GO Brazil, Greece, Canada, Iraq, and South Africa [6,28,34–37]. The potential explanation for this association is that mothers who have breastfed before are more familiar with the process and may have greater confidence in their ability to breastfeed. This aligns with Bandura's theory of breastfeeding self-efficacy, which suggests that personal experience enhances one's belief in their capability to perform a specific activity [38].

Regarding pregnancy status, this study revealed that women who had planned their pregnancy were four times more likely to have a high BFSE status compared to those who had not planned their pregnancy. This finding is supported by studies conducted in Turkey and Tabriz, Iran [39,40]. The strong desire to become pregnant may contribute to a positive mindset toward pregnancy and increase the likelihood of having a high BFSE status. However, a study conducted in Santo André, São Paulo, Brazil, did not find a significant association between pregnancy status and breastfeeding self-efficacy [24] which contrasts with our findings. A possible justification for this might be that the study used probability sampling techniques, which may limit our understanding of the population's reality. Educating postpartum mothers to enhance their knowledge and foster positive attitudes toward breastfeeding may improve breastfeeding self-efficacy (BFSE) and lead to better breastfeeding practices among women in Ethiopia. Improving breastfeeding self-efficacy, particularly in culturally diverse populations, requires a nuanced approach that respects and incorporates cultural beliefs, practices, and social norms. Therefore, implementing culturally tailored solutions that

can improve breastfeeding self-efficacy by fostering an environment that values and supports breastfeeding within the context of the mother's cultural identity is recommended.

## Strength and limitation

This study utilized multiple variables to examine factors related to breastfeeding self-efficacy. However, due to its cross-sectional design, it is limited in establishing a cause-and-effect relationship. This study also has a limitation in that the data collected from the mothers' interviews and questionnaires was based only on their self-reports, therefore we cannot exclude the possibility that mothers in the study might have exhibited recall bias and higher social desirability to provide the correct answer to the questions asked.

## Conclusion

Overall, in this study, it was found that more than half of postpartum mothers have a low BFSE status. The current study also found that mothers' age, mothers' education, mode of delivery, and breastfeeding experience were predictors of BFSE. Based on the results of this study, the following suggestions are provided: there is a need for increased focus on improving the educational status of mothers, mothers who give birth by C-section should receive additional counseling and support from healthcare providers to enhance their BFSE status before being discharged from the hospital. Additionally, the study recommends that further research be conducted at the community level.

## Supporting information

**S1 Questionnaire. Questionnaires (English and Amharic versions)**
(ZIP)

**S2 Data. Data set**
(XLS)

## Acknowledgment

We would like to thank Jimma University and we appreciate the study's participants for agreeing to participate in this study.

## Author contributions

**Conceptualization:** Lemlem Nigussiee, Tigist Demeke, Vinod Bagilkar, Yemisrach Firetsidk, Mulatu Abageda, Tefera Belachew.

**Data curation:** Lemlem Nigussiee.

**Formal analysis:** Lemlem Nigussiee, Tigist Demeke.

**Funding acquisition:** Lemlem Nigussiee.

**Investigation:** Lemlem Nigussiee, Tigist Demeke, Vinod Bagilkar, Yemisrach Firetsidk.

**Methodology:** Lemlem Nigussiee, Tigist Demeke, Vinod Bagilkar, Yemisrach Firetsidk, Tefera Belachew.

**Resources:** Lemlem Nigussiee.

**Software:** Lemlem Nigussiee, Tigist Demeke, Yemisrach Firetsidk, Mulatu Abageda.

**Supervision:** Lemlem Nigussiee, Tigist Demeke, Vinod Bagilkar, Tefera Belachew.

**Writing – original draft:** Lemlem Nigussiee, Tigist Demeke, Vinod Bagilkar, Yemisrach Firetsidk, Mulatu Abageda, Tefera Belachew.

**Writing – review & editing:** Lemlem Nigussiee, Tigist Demeke, Vinod Bagilkar, Yemisrach Firetsidk, Mulatu Abageda, Tefera Belachew.

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
