## [Decision Letter · Decision Letter 0]

2 Jan 2024

PONE-D-23-34434Breastfeeding self-efficacy status and associated factors among postpartum mothers at Hadiya Zone public hospitals, southern EthiopiaPLOS ONE

Dear Dr. Abageda,

Thank you for submitting your manuscript to PLOS ONE. After careful consideration, we feel that it has merit but does not fully meet PLOS ONE’s publication criteria as it currently stands. Therefore, we invite you to submit a revised version of the manuscript that addresses the points raised during the review process.

We look forward to receiving your revised manuscript.

Kind regards,

Nega Degefa Megersa, Msc

Academic Editor

PLOS ONE

Journal Requirements:

"We would like to thank Jimma University for providing the financial support necessary to carry out this research. We appreciate the study's participants for agreeing to participate."

"This study received a student research fund from Jimma University. The funder has not played any role in the study design, collection of data, or development of this manuscript."

6. Please clarify the number of Tables uploaded in your manuscript and PDF file. 

Additional Editor Comments:

Adding line numbers to the document would facilitate easier and more precise annotation.

Background

The background section should thoroughly address the problem's context, identify the existing research gap, and establish the significance of the study. The current paper, however, falls short in adequately addressing these aspects.

Method:

What factors are considered when selecting candidate variables for multivariable logistic regression analysis, and why is a p-value cutoff of 0.25 used? Additionally, are there any other criteria that should be considered when selecting candidate variables?

I would appreciate more information about the 416 individuals who participated in this study. I'm curious to learn how you selected the four hospitals for this study from among those available in the zone.

Which specific measure of association was employed to quantify the association between the variables of interest?

When determining the sample size, it is essential to consider both the statistically significant factors from prior research and the single population proportion formula. The larger of the two calculated sample sizes should be adopted to ensure sufficient statistical power for the study.

Result

The inclusion of the confidence interval alongside the prevalence of high-level BFSE enhances the precision and reliability of the findings, offering a more accurate reflection of the prevalence in the target population.

It would be beneficial to ensure consistency in the use of decimal points throughout the text. For instance, the first AOR value is presented with three decimal places, the second with four, and the last with only one, which is inconsistent. To maintain consistency, the AOR values and their corresponding confidence intervals should be formatted with the same number of decimal places.

Conclusion

The conclusion fails to provide a comprehensive interpretation of the study's main result and does not align with the existing body of knowledge. It also neglects to discuss the factors associated with BFSE and the implications of the research findings.

Discussion

Needs a complete revision. The work of discussion must reveal the relative importance of the current findings. Implications of the findings, discussion of the limitations of the study, and guidance on future directions for further research.

Reviewers' comments:

Reviewer's Responses to Questions

**Comments to the Author**

1. Is the manuscript technically sound, and do the data support the conclusions?

Reviewer #1: Yes

Reviewer #2: Yes

Reviewer #3: Yes

Reviewer #4: Partly

2. Has the statistical analysis been performed appropriately and rigorously? 

Reviewer #1: Yes

Reviewer #2: Yes

Reviewer #3: Yes

Reviewer #4: Yes

3. Have the authors made all data underlying the findings in their manuscript fully available?

Reviewer #1: Yes

Reviewer #2: No

Reviewer #3: Yes

Reviewer #4: Yes

4. Is the manuscript presented in an intelligible fashion and written in standard English?

Reviewer #1: Yes

Reviewer #2: No

Reviewer #3: Yes

Reviewer #4: No

5. Review Comments to the Author

Reviewer #1: Review comments

Thank you dear editors for you’re considering me to review this paper!

I have completed my evaluation of the manuscript with the Manuscript Number: PONE-D-23-34434 entitled as “Breastfeeding self-efficacy status and associated factors among postpartum mothers at Hadiya Zone public hospitals, southern Ethiopia”

It is an interesting topic and well written paper. However, I have the following comments to the authors for improvement:

Abstract:

1) In the result section you wrote “secondary and above educational status [AOR = 6.856; 95% CI (2.670-17.603)]” but in your model table it said “More than secondary” consistency problem

2) It is not clear “statistical package for social science on Windows version 24.”

Background:

3. Put a reference to this sentence ““If self-efficacy is lacking, people tend to behave ineffectually, even though they know what to do’’ says Albert Bandura.”"

4. put references to this sentence “Finally, breastfeeding self-efficacy status is influenced by sociodemographic, obstetric, and health service-related variables and has not been researched before in postpartum mothers in Ethiopia.”

Methodology:

5. Why use p=50% and single population proportion formula, why not double population proportion?

6. How the samples were allocated to four study hospitals?

7. Did you show us Multicolinarity results?

Result:

Discussion:

8. It needs more revision especially “48% of mothers had a high BFSE status; this result is higher than that of a study carried out in Brazil, where 35% of mothers had high BFSE (24), 43.9% of respondents in China had high BFSE (25), and 43.2% of respondents in Indonesia had high BFSE [18]. The reason might be because of sociocultural and economic variation among the countries. This might be the possible reason for the discrepancy in mothers’ BFSE status.” How we can identify your result is higher than the listed references? Put the 95%CI first?

9. “Nepal, which was 94% [26]. This variation may be due to the smaller sample size, and the sample was collected after the mothers had been breastfeeding for at least two weeks. Involvement in BF or practice increases the possibility of having a higher status in BFSE [27]. Moreover, in Brazil, 95.3 percent of postpartum mothers have a high BFSE status (28). This variation might be secondary to the purposive sampling technique used and the way of classifying the status of BFSE being different, which was categorized into three.” It is unclear which study has small sample size, use purposive sampling technioque; the current or previous one and the like…..?

10. Put a reference to this paragraph “The possible justification might be that elderly women are more likely than younger women to have several children. As a result, this presents a chance to breastfeed repeatedly. This shows that having prior BF experience results in having a high status of BFSE.”

11. “This outcome is supported by research conducted in western Indonesia [18, 32, 33]” are all reference from Indonenia?

Generally the discussion part needs more clarity and rewriting.

Conclusion:

It lacks recommendation.

Your model table is 1/5?

I hope the authors will be properly address the questions.

Finally thank you!!

Reviewer #2: Thanks for inviting me to review the paper.

I read the paper in detail. It was good but still there are numerous points to be revised.

A.Abstract

Results

A p-value of <!--=0.05 and 0.25 were used for multivariate and bivariate analysis. But at 0.05 there is no association. Do think this is correct. <br /

1.Introduction.

1.1 There is many grammatical, chronological problems. There is sentences which are not related but combined together.

1.2 Only two sentences said about BFSE. 95% of the Introduction part is about EBF.

1.3 in paragraph 3 it said 166 million of Diarrhoea and 9 million pneumonia caused by BF. What is it mean that?

1.4 You said about factors associated with EBF instead of factors associated with BfSE.

1.5 Gap with previous studies not well addressed.

1.6 Over all the introduction's part needed to be revised .

2.Methods

2.1a. Exclusion criteria is not clear

what if women gave birth home and referred due to complications.

2.2 you excluded those did will to participate but in you result you included by said 98% of response rate. Secondly you already add 10% of non response rate therefore Exclusion of such criteria is not acceptable.

2.3 What about those mothers who severely ill?

2.4. Sample size determination.

There are many studies conducted in the same topics but you take 59% to calculate, could you please justify?

2.5 Sampling technique and procedure

It better if you could demonstrate how you proportionally distributed to each hospital.

2. 6 Again some hospital are comprehensive other are general and primary. How you easily compare those hospital with different services.

2.7 Data collection process

Who were your data collectors?

2.8 Do you think that you check the validity and reliability of the questionnaire, if so please elaborate accordingly.

2.9 You are expected to write the questionnaire used to assess BF SE.

2.10 Could you please respond on why media was used? Why not mean? Please justify?

2.11 Analysis: at p-value 0.05 there is no association. Unfortunately you used it as significantly associated.

2.12 You paper lacks the quality control parts.

3. Results

3.1 proportion of BFSE was not illustrated in the results?

3.2 Don't you think that parity and BF experience has multicolinarity?

3.3 Factors associated with BFSE .you reported only age and mode of delivery why not others.

4. Discussion

4.1 Please revise it again

4.2 All lot of grammatical and chronological problems.

4.3. Your study lack limitations and strengths

5.Ethical consideration for young less than 18 years need to be addressed

Reviewer #3: General

• It would be better for the authors to use line number.

• The study appears to be good, but in order for the paper to be readable, the authors should check the language /grammar throughout the document. And also revise for redundancy.

Background

• On the first statement, it is not the correct way of writing term definition including the punctuation

• On the 3rd paragraph, the Authors should rewrite the following statement; “This shows that a high breastfeeding self-efficacy status has power in breastfeeding, whereas it is a reason for failure to BF if it goes lower [3–5].”

• The authors’ use of abbreviation and a full term should be consistent throughout the document i.e. the word breastfeeding and BF

• On the 4th paragraph, 2nd line, change the words “children under five” to under five children.

• On the 4th paragraph, 2nd statement, what does the statement indicate? It is stating breast feeding causes illnesses. “However, based on statistics from 130 nations' health surveillance, an additional 166 million episodes of diarrheal illnesses and 9 million instances of pneumonia in children under the age of two were caused by BF”.

The authors should revise and rewrite

• Better to rewrite the fourth paragraph totally.

Method

• The authors should rewrite the statement under the population subtitle and make it clear. They have stated that the source population for the study was postpartum mothers at selected hospitals in the zone, but it also states that all hospitals with in the zone are included in the study.

• On the sampling method, there is unnecessary detail about the non-response rate. “To compensate the participants who refused or stopped the interview before completion of the questionnaire after they were selected to participate in the study”

• The authors need to clarify how systematic random sampling technique was implemented for postpartum mothers with in 24hours? It is also known that mothers who gave birth spontaneous vaginal delivery stays mostly for 6 hours at hospitals.

• On the data analysis, line 5, it is better to use the term binary logistic regression rather than bivariate since there is a difference between the two. Including on the table caption

• On the data analysis, line 7, the authors should not use the word predictors since the study was cross sectional. Including on the table caption

Result

• While the result was well written, one of the descriptive results under the subtitle obstetric and health service related factors is not in line with result with in the table. Specifically, line 1, which states 72.6% and the table states 75 %

• Also the table number in descriptive result and the table at the end are not in line with each other.

Discussion

• On the first and second paragraph a study from Brazil is mentioned with different percentage. The authors should specify the study settings.

• In the second paragraph, second statement, it is better to find another reason for variation among studies rather than small sample size.

• On the second paragraph, line 6, the authors should clarify how a purposive sampling technique can cause a variation among results.

• Rewrite the comparison with the study done in Iran. And also the study participants are not similar (pregnant vs. postpartum)

Conclusion

• It is better to rewrite the conclusion

• It is better to include clinical implication and recommendation in line with the result.

Tables

• The author should check the table numbering and order

• The table on page 19 should also include the variables that were significant in the bivariable regression but not in the multivariable one. i.e. ANC care visit, parity, etc.

Reviewer #4: Reviewers’ response to editor and author’s comments

Title: Breastfeeding self-efficacy status and associated factors among postpartum mothers at Hadiya Zone public hospitals, southern Ethiopia

Manuscript ID: PONE-D-23-34434

From: Reviewer

To: The editor in chief, PLOS ONE

My sincere gratitude will go out to you for sharing this type of research with the scientific community. It has been a pleasure learning about it.

Background: In general, it should be revised because of poor construction of statements, flow and it can’t convince readers so it needs a thorough revision)

References should not be put in square brackets.

What is the difference between a facility and a health institution?

Why you want to conduct in the selected health facilities

The educational status confidence interval is a bit wider, why?

Method section: sampling method, an incomplete kind of phrase. So, you need to fully write it as sample size determination and sampling Procedure. Again, you ought to write as a data collection tool and procedure.

To obtain insights, this research must be conducted on a community level, as we must take into account cultural and contextual factors.

Please write formally or follow the journal guideline (data collection) like……Data collection tool and procedure.

Who collected your data?

Who were your supervisors?

Did you conduct a pretest? Nothing is mentioned

Again, you fail to narrate the internal consistency of the data collection tool (i.e. reliability).

I didn’t get, how you control /assure the quality of the data.

As you described in the data processing and analysis section, multicollinearity was checked, how did you check it, and what was the result and interpretation, is any variable that scored above acceptable? if any, please …..

Why did you prefer to use p<0.25? why not p<0.20?

Where is the operational definition? We don’t know how you measure it.

Since you have an ordinal dependent variable i.e. the level of breastfeeding efficacy, ordinal logistic regression????? Why dichotomy it?

What is the relationship between breastfeeding self-efficacy and breastfeeding technique?

Why do you want to conduct only public hospitals?

why do you say low the proportion of BFE in your conclusion? What are your comparators? You mention the standard.

Why did you consider P is 50%?

Where is the sampling procedure? I didn’t find even a reference for it.

Result

Please write a full statement of the result section

Avoid using the word “study subject” better to use study participants

The standard Deviation is 5.32 which means there is a data set variability that could be outliers /extreme values/ or other factors therefore I suggest you first check the age distribution whether skewed or not or you might consider transformation or you might conduct a sub-group analysis.

Please summarize it.” Mothers were asked about the educational status of their husbands”

Please merge (put in one table together) Table 1 and Table 2

I am not clear about the result of breastfeeding efficacy because you mentioned in your document to measure it you considered the median but here you mentioned taking a single question from the table so there is incompatibility in answering the dependent variable.

What do you mean by “Bivariable analysis in the binary logistic regression model”?

What is the valid reason for the association between BFE with spontaneous delivery?

What are the strengths and limitations of your study?

Discussion

There is a resulting mismatch of 35.6% vs 48% you have to correct it. (there is inconsistency /result incompatibility so it is a major comment).

Did you check Brazil, Indonesia, and China match their inclusion and exclusion criteria?

Frequently you mention “Respondents” Please use other terms or minimize them.

Despite this research is indispensable and contributes to the body of knowledge, your way of writing needs more work. The paper will benefit from professional proofreading and editing to

improve the clarity of the written language

6. PLOS authors have the option to publish the peer review history of their article (what does this mean? ). If published, this will include your full peer review and any attached files.

**Do you want your identity to be public for this peer review?** For information about this choice, including consent withdrawal, please see our Privacy Policy .

Reviewer #1: No

Reviewer #2: No

Reviewer #3: No

Reviewer #4: No

---

## [Author Response · Author response to Decision Letter 0]

20 Feb 2024

We have attached a point-by-point response to the reviewer and editor comments.

---

## [Decision Letter · Decision Letter 1]

15 Nov 2024

PONE-D-23-34434R1Breastfeeding self-efficacy status and associated factors among postpartum mothers at Hadiya Zone public hospitals, southern EthiopiaPLOS ONE

Dear Dr. Abageda,

Thank you for submitting your manuscript to PLOS ONE. After careful consideration, we feel that it has merit but does not fully meet PLOS ONE’s publication criteria as it currently stands. Therefore, we invite you to submit a revised version of the manuscript that addresses the points raised during the review process.

We look forward to receiving your revised manuscript.

Kind regards,

Ammal Mokhtar Metwally, Ph.D (MD)

Academic Editor

PLOS ONE

Journal Requirements:

Additional Editor Comments (if provided):

This study is scientifically significant, offering valuable insights into the factors influencing breastfeeding self-efficacy in Ethiopian postpartum mothers. It provides a foundation for future research and practical recommendations for public health programs. With some improvements in methodological rigor and deeper exploration of unique cultural contexts, the study could have a stronger impact on the field. The writing is generally clear, but it could be improved by reducing redundancy and providing more specific methodological details.

Please consider the following:

1) The study makes a valuable contribution to understanding breastfeeding self-efficacy and its influencing factors in Ethiopia, but it could push the boundaries of knowledge further by exploring new interventions or cultural influences in more depth.

2) The self-reported data in the methodology section introduces potential biases (e.g., recall or social desirability bias), which weakens the reliability of the findings. There is no mention of how data quality was ensured or validated.

3) There is insufficient detail on the statistical techniques, handling of missing data, or variable selection in the analysis. A more transparent explanation would improve confidence in the findings.

4) While the study highlights important socio-demographic factors, it could provide more concrete suggestions for intervention strategies or programmatic changes in healthcare settings.

5) The study is significant but could have a larger impact if it offered more innovative or culturally-tailored solutions in the discussion section to improving BFSE.

Reviewers' comments:

Reviewer's Responses to Questions

**Comments to the Author**

1. If the authors have adequately addressed your comments raised in a previous round of review and you feel that this manuscript is now acceptable for publication, you may indicate that here to bypass the “Comments to the Author” section, enter your conflict of interest statement in the “Confidential to Editor” section, and submit your "Accept" recommendation.

Reviewer #1: All comments have been addressed

2. Is the manuscript technically sound, and do the data support the conclusions?

Reviewer #1: Yes

3. Has the statistical analysis been performed appropriately and rigorously? 

Reviewer #1: Yes

4. Have the authors made all data underlying the findings in their manuscript fully available?

Reviewer #1: Yes

5. Is the manuscript presented in an intelligible fashion and written in standard English?

Reviewer #1: Yes

6. Review Comments to the Author

Reviewer #1: All comments are addressed; no needs of additional comments left to be address by authors. the paper is accepted as it is

7. PLOS authors have the option to publish the peer review history of their article (what does this mean? ). If published, this will include your full peer review and any attached files.

**Do you want your identity to be public for this peer review?** For information about this choice, including consent withdrawal, please see our Privacy Policy .

Reviewer #1: No

---

## [Editor Report · Decision Letter 2]

6 Jan 2025

Breastfeeding self-efficacy status and associated factors among postpartum mothers at Hadiya Zone public hospitals, southern Ethiopia

PONE-D-23-34434R2

Dear Dr. Abageda,

We’re pleased to inform you that your manuscript has been judged scientifically suitable for publication and will be formally accepted for publication once it meets all outstanding technical requirements.

Kind regards,

Ammal Mokhtar Metwally, Ph.D (MD)

Academic Editor

PLOS ONE
---

## [Editor Report · Acceptance letter]

PONE-D-23-34434R2

PLOS ONE

Dear Dr. Abageda,

I'm pleased to inform you that your manuscript has been deemed suitable for publication in PLOS ONE. Congratulations! Your manuscript is now being handed over to our production team.

Kind regards,

on behalf of

Professor Ammal Mokhtar Metwally

Academic Editor

PLOS ONE